# In their own words: An Australian community sample's priority concerns regarding mental health in the context of COVID-19

**Marlee Bower\***, **Amarina Donohoe-Bales**, **Scarlett Smout**, **Andre Quan Ho Ngyuen**, **Julia Boyle**, **Emma Barrett**, **Maree Teesson**

The Matilda Centre for Research in Mental Health and Substance Use, The University of Sydney, Sydney, New South Wales, Australia

\* marlee.bower@sydney.edu.au

**Data Availability Statement:** Due to legal and ethical restrictions imposed by the Human Research Ethics Committee (University of Sydney)

## Abstract

The COVID-19 pandemic has resulted in significant and unprecedented mental health impacts in Australia. However, there is a paucity of research directly asking Australian community members about their mental health experiences, and what they perceive to be the most important mental health issues in the context of the pandemic. This study utilises qualitative data from *Alone Together*, a longitudinal mixed-methods study investigating the effects of COVID-19 on mental health in an Australian community sample (N = 2,056). A total of 1,037 participants, ranging in sex (69.9% female), age (M = 40–49 years), state/territory of residence, and socioeconomic status, shared responses to two open-ended questions in the first follow up survey regarding their mental health experiences and priorities during COVID-19. Responses were analysed using thematic analysis. Participants described COVID-19 as primarily impacting their mental health through the disruption it posed to their social world and financial stability. A key concern for participants who reported having poor mental health was the existence of multiple competing barriers to accessing high quality mental health care. According to participant responses, the pandemic placed additional pressures on an already over-burdened mental health service system, leaving many without timely, appropriate support. Absent or stigmatising rhetoric around mental health, at both a political and community level, also prevented participants from seeking help. Insights gained from the present research provide opportunities for policymakers and health practitioners to draw on the expertise of Australians' lived experience and address priority issues through targeted policy planning. This could ultimately support a more responsive, integrated, and effective mental health system, during and beyond the COVID-19 pandemic.

## Introduction

The COVID-19 pandemic and ensuing social and economic disruption has resulted in unprecedented challenges for mental health and wellbeing worldwide [1,2]. Emerging Australian research has shown that widespread infection control measures including physical distancing,

surrounding sensitive data and participant privacy concerns, we are unable to make the full dataset publicly available. In accordance with PLOS ONE 'guidelines for qualitative data', participants did not consent to have their full excerpts made publicly available. Additionally, full excerpts contain confidential and potentially identifiable information. Access to de-identified data can be made available upon reasonable request by contacting any of the authors via alone-together.study@sydney.edu.au or the University of Sydney Ethics department at human.ethics@sydney.edu.au.

**Funding:** This work was supported by the Henry Halloran Trust under a Festival of Urbanism 2020 Grant (MB): https://www.sydney.edu.au/henry-halloran-trust/ This work was also supported by the LifeSpan Research Network, Faculty of Medicine and Health at the University of Sydney Seed Funding Grant (MB): https://www.sydney.edu.au/medicine-health/our-research/research-centres/lifespan-research-network.html The funders had no role in study design, data collection and analysis, decision to publish, or preparation of the manuscript.

**Competing interests:** The authors have declared that no competing interests exist.

self-isolation and localised lockdowns may have significant, long-lasting impacts on psychological health. A cross-sectional Australian study found more than half of participants reported a deterioration of mental health during the initial COVID-19 lockdown period in 2020 [3]. Trends in poor mental health have persisted for over a year since Australia's first COVID-19 case, with approximately one in five Australians reporting their mental health as 'worse' or 'much worse' than prior to the pandemic [4].

Despite this growing evidence and the serious disruption COVID-19 has caused in the daily lives of Australians, there is a paucity of research directly asking Australian community members about their experiences. In particular, research identifying general population mental health priority concerns in Australia during the COVID-19 pandemic is scarce. Public policy informed by the voices of the general population, including those with and without lived experience of mental ill-health, is critical to facilitate more responsive and targeted investment to meet the mental health needs of Australians during and beyond the pandemic [5].

Much of the pre-pandemic research examining the mental health experiences and concerns of Australians has been conducted primarily with consumers of mental health services and their carers. These studies have broadly identified several areas of concern around service provision, such as inaccessibility and low-quality of mental health services; high thresholds of illness required to access treatment; unavailability of culturally appropriate services; poor consideration of holistic consumer needs; high cost of services and concerns around confidentiality or ineffectiveness of treatment [6–9]. These issues may have been exacerbated by the COVID-19 pandemic, as new evidence suggests that Australian mental health consumers have disengaged from psychosocial support services due to COVID-related delays or disruptions to service provision, despite a growing need for mental healthcare [10].

Another prominent issue reported amongst mental health consumers and carers pre-pandemic is the social stigma and discrimination associated with having mental health issues. Reavley and Jorm [11] previously found that stigma and discrimination contributed to consumers' experiences of distress and adversely impacted their relationships. Other research with mental health consumers and carers found discriminatory interactions with healthcare workers led to negative self-esteem and heightened substance use [12,13]. Self-stigma in the form of negative public perceptions and attitudes around professional help-seeking, as well as personal beliefs that the individual can or should address their mental health issues on their own, also acted as a barrier to accessing and engaging necessary care [6,11].

Most of the aforementioned studies were conducted amongst consumer and carer populations prior to COVID-19. However, there exists a significant knowledge gap around the mental health experiences of the general population, and non-consumers, during the pandemic. Research in the COVID-19 context would benefit from including the voices of those outside the mental health service system for several reasons. First, existing barriers to mental health care and services contribute to unmet mental health needs in the general population, indicating that not all people experiencing mental illness and distress are consumers [14]. Next, COVID-19 has been a major disruption to our social and economic fabric, and has subsequently provided a society-wide shock to the mental health of Australians [4]. Researchers have highlighted the importance of increasing mental health service capacity in Australia due to high levels of COVID-related distress in the general population, with telephone crisis counselling service Lifeline recently recording its highest daily number of calls in almost 60 years of operation [15,16].

Existing research has also been primarily quantitative. However, qualitative research is uniquely placed to provide insight into perceived barriers to care and other existing issues surrounding mental health. Garnering perspectives from a variety of people, including those with and without direct experience of the Australian mental health system, can allow for a rich and

wide-angled perspective on what major factors in Australian society support peoples' mental health, as well as identifying existing social and systems-level barriers to achieving mental health in the COVID-19 context. Therefore, this paper aims to explore Australian community members perspectives on mental health priorities during the pandemic by drawing on qualitative data from a diverse community sample of Australian adults across all states and territories. Based on pre-pandemic and emerging evidence, it is hypothesised that participants will report issues and priorities around COVID-specific social and economic changes on mental health and wellbeing and related impacts on mental health service access and provision.

## Material and methods

Data included in the present study were collected as part of *Alone Together*: a mixed-methods, longitudinal study investigating the mental health and substance use impacts of COVID-19-related changes to Australian general community members' lives, with a particular focus on the role of the social determinants of health [17]. Full ethics approval was obtained from the University of Sydney's Human Research Ethics Committee (HREC; 2020/460). An active consent procedure was used. Participants were provided with a participant information statement and then asked to confirm their consent before they were able to commence the questionnaire. In addition, the steps to withdraw from the study at any time were clearly explained. Prior to commencing each follow-up questionnaire, participants were also asked to actively re-confirm their consent to take part in the study.

During the baseline survey period between July and December 2020, participants across all Australian states and territories were recruited through targeted advertisements on social media (i.e., Facebook, Instagram, Twitter), community websites, and physical flyers in public spaces. Participants were also recruited through charitable organisations providing support, social care, accommodation or health services for those experiencing, or at risk of, homelessness, social isolation and financial disadvantage, to reach a broad sample of Australians, including those unlikely to use social media. In these instances, researchers would display flyers or discuss the research process and ethics with prospective participants. To be eligible to participate in the *Alone Together* study, participants were required to be 18 years or older, currently residing in Australia and have sufficient English proficiency to understand the survey and consent procedures.

The baseline survey included questions pertaining to participant demographics, such as gender, age, income, employment, housing characteristics, residence and marital status. Participants who completed the baseline survey (N = 2,056) were invited to participate in a follow-up survey between March and June 2021 via email, calls and/or text messages. Interested participants were directed to Qualtrics, a secure online survey platform, to complete the 30–60-minute survey. Participants originally recruited through housing/homelessness service organisations were given the opportunity to take part in a face-to-face follow-up survey, conducted at the organisation's premises using hardcopy surveys and visual showcards to increase accessibility for participants with low literacy. A total of 1,350 participants (65.66% of the baseline sample) completed the first follow-up survey.

As part of the first follow-up survey, participants were asked two optional open-ended questions: 'What do you think are the most important issues around mental health in Australia today?' and 'What impact has COVID-19 had on your mental health, emotions, and/or wellbeing in the past 6 months?'. Both questions stated 'answer with as much or as little detail as you wish'.

Using Braun and Clarke's Thematic Analysis methodology [18], data were de-identified and inductively coded by two researchers independently (MB, ADB), to ensure the highest

level of data quality control, applying brief names and descriptions to each unique concept that emerged from the data to identify the ideas expressed by each participant [19]. Coded data were then compared, and the coding framework given minor alterations to reflect the findings of both researchers. NVivo, a qualitative data management programme, was used to facilitate coding using this combined framework [20]. Similar codes were grouped together under broad, high-level categories and connections were identified between codes. Coded data were iteratively read and analysed amongst the research team, who collaboratively and unanimously arrived at two primary themes through iterative discussion.

Participants who responded to at least one of the two questions of interest were included in this study (n = 1,037). Approximately 80% of those who completed the follow-up survey answered at least one question (N = 1,037 total: n = 972 for the first question and n = 1008 for the second question).

## Results

### Participant demographic characteristics

The sample ranged from age 18–89 and the median age bracket was 40–49 years;69.9% (n = 725) identified as female, 7.7% spoke English as a second language (n = 78), and 0.6% (n = 6) were Aboriginal and/or Torres Strait Islander. The median weekly income bracket of the sample was $1075-$1700 (n = 198) and ranged from less than $300 per week (n = 47) to more than $2400 per week (n = 228). Further participant characteristics are presented in Table 1, compared against demographics of the general Australian population.

### Qualitative findings

Participants described several everyday factors as being important to their mental health and/ or the mental health of other Australians. They described risk factors including lack of secure

**Table 1. Participant characteristics at baseline (N = 1037).**

|  | % (n) | Australian general population |
|---|---|---|
| **Employment Status** |  |  |
| Full-time or Part-time | 49.4% (512) | 56.1%[a] |
| Casual or Sub-contractor | 7.8% (81) |  |
| Unemployed | 6.0% (62) | 4.1%[a] |
| Student | 8.9% (92) | Not available |
| Retired | 15.6% (162) | 33.1%[a] |
| Other (homemaker, volunteer) | 12.4% (128) |  |
| **Marital Status (Married)** | 56.6% (587) | 48.1%[b] |
| **Residency** |  |  |
| New South Wales | 43.5% (451) | 32.0% [c] |
| Victoria | 31.8% (330) | 26.2%[c] |
| Queensland | 9.4% (97) | 19.8%[c] |
| Western Australia | 4.5% (47) | 10.2%[c] |
| Australian Capital Territory | 4.1% (42) | 1.7%[c] |
| South Australia | 4.1% (42) | 7.0%[c] |
| Tasmania | 2.1% (22) | 2.1%[c] |
| Northern Territory | 0.6% (6) | 0.9%[c] |
| **Live in Major City** | 74.5% (702) | 72.3%[d] |

*(Continued)*

**Table 1.**  (Continued)

| | % (n) | Australian general population |
|---|---|---|
| **Live in Inner Regional Area** | 16.9% (159) | 17.7% [d] |
| **Live in Outer Regional or Remote Area** | 8.6% (81) | 9.9% [d] |
| **Housing** | | |
| Own their property | 54.0% (522) | 66% [e] |
| Renting | 37.9% (266) | 32% [e] |
| Other housing arrangement (e.g.,) | 8.1% (79) | 2% [e] |
| **Household's Main Source of Income** | | |
| Wage or Salary | 69.1% (716) | 75% [f] |
| Own Unincorporated Business | 1.5% (15) | 5% [f] |
| Government Allowance or Pension | 16.9% (175) | 8% [f] |
| Superannuation or Annuity | 9.4% (98) | 12% [f] |
| Other | 3.2% (33) | |
| **Money available for Expenses (after cost of housing)** | | Not available |
| Essential expenditure (e.g., bills, transport, and food) | 93.1% (965) | |
| Non-essential Expenditure (e.g., social activities or holidays) | 78.2% (811) | |
| Savings or Investment | 58.3% (605) | |
| **Mental Health Status** | | |
| Previously diagnosed mental health disorder | 48.7% (471) | 45.5% [g] |
| Seen GP for mental health concerns in past 6 months | 39.9% (414) | Not available |
| Seen Psychologist for mental health concerns in past 6 months | 27.7% (287) | |

[a] ABS data from 2016 Census of population and housing. Employment status for persons aged 15 years and over. https://www.abs.gov.au/AUSSTATS/abs@.nsf/DetailsPage/2071.02016?OpenDocument.

[b] ABS data from 2016 Census of registered marital status of people in Australia aged 15 years and over. https://quickstats.censusdata.abs.gov.au/census_services/getproduct/census/2016/quickstat/036.

[c] ABS data of national, state and territory population statistics, March 2021. https://www.abs.gov.au/statistics/people/population/national-state-and-territory-population/latest-release.

[d] The Australian Bureau of Statistics Remoteness Areas, which divides Australia into five levels of remoteness based on relative distance and access to services, was used to convert participant postcodes into levels of remoteness. https://www.abs.gov.au/websitedbs/d3310114.nsf/home/remoteness+structure.

[e] ABS data from 2017–18 Survey of Income and Housing. Housing tenure status for Australian households. https://www.abs.gov.au/statistics/people/housing.

[f] Australia's population by main source of household income derived from Australian Council of Social Service (ACOSS). https://povertyandinequality.acoss.org.au/inequality/australias-population-by-main-source-of-household-income-2016/.

[g] Australian Government Department of Health data on the prevalence rate of mental disorder lifetime diagnosis derived from the 2007 National Survey of Mental Health and Wellbeing. https://www1.health.gov.au/internet/publications/publishing.nsf/Content/mental-pubs-m-mhaust2-toc~mental-pubs-m-mhaust2-hig~mental-pubs-m-mhaust2-hig-pre.

income, meaningful employment and social connections; and barriers to improving their mental health, including difficulty accessing adequate and appropriate mental health care and stigmatising and/or absent rhetoric around seeking help for mental health issues within an Australian context. Broadly, the pandemic was described as exacerbating both risk factors and barriers to care (Fig 1). The following section includes de-identified participant quotes to illustrate broader findings and support underlying themes.

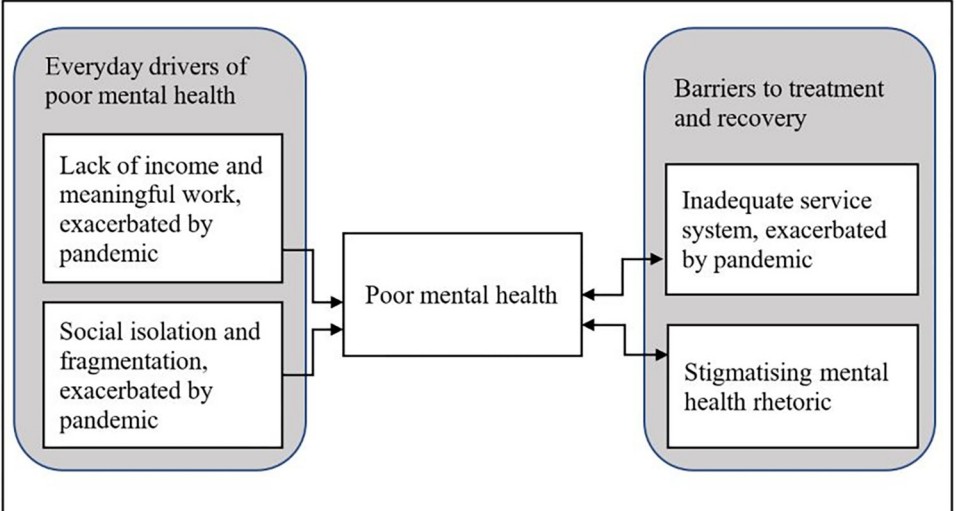

**Fig 1. Key themes exacerbated during the COVID-19 pandemic—Everyday drivers of poor mental health and barriers to treatment and recovery.**

**Money, work and relationships: COVID-19 exacerbated everyday drivers of poor mental health.**   While some participants described COVID-19 as creating some new and unique mental health concerns, including heightened feelings of *'uncertainty'* and *'apprehension'* about the future *(n = 51)*, participants more often reported that COVID-19 had *'exacerbated'* and *'intensified'* existing personal and systemic mental health issues. Approximately one fifth (n = 236) of participants' accounts reflected two drivers through which the pandemic impacted Australian community members' mental health: a) increases in financial hardship and precarity, and b), changes in the participant's social world, relationships and supports. While the two drivers will be described separately, it is important to note that participants recognised the importance and interconnectedness of these broader factors and their conjoint influence on mental health and wellbeing, as is evident in the following participants' accounts:

*'[I have experienced] ongoing psychological challenges leading to depression and anxiety', based on 'COVID-19 and it's various impacts. . .[including] social isolation, financial difficulties, and loss of work.' (Male, 46, Victoria)*

*'Pandemic conditions, including isolation, limited exercise, increased uncertainty, job loss, lack of purpose, easily escalated mental health problems.' (Female, 29, NSW).*

*Stress associated with employment and income.* Participants (n = 56) reported experiencing *'fear'*, *'stress'*, *'anxiety'* and *'hopelessness'* associated with *'precarious'* employment, underemployment, or unemployment, often related to the pandemic. Many participants described *'job uncertainty'*, *'casualisation of the workforce'*, *'loss of job security'* and being *'fearful'* about supporting one's family, *'paying bills'* and *'keeping a roof over one's head.'*

Broadly, when participants talked about the impact of lost work on their mental health, they described the loss of their place within broader social structures and loss of the sense of *'meaning'*, *'confidence'* and *'engagement'* that work had previously provided. Participant accounts showed that becoming unemployed had negatively shifted the way they felt about themselves, with many reporting feelings of *'lower hope and self-esteem'* and *'self-respect'*. Sometimes this lower self-esteem was linked to stigmatising government rhetoric around

people who are unemployed: *'the government does not see that mental impact [of] being unemployed and getting the distinct feeling you are seen as scum' (Female, 39, NSW)*.

The loss of income associated with the pandemic (n = 42) and lost work was also described as causing stress, particularly by limiting some participants' ability to pay for increasing housing costs. Some participants (n = 36) identified housing stress as a key priority for their mental health, particularly *'unaffordable housing prices' (Female, 30, NSW)*, the *'inaccessibility of the housing market' (Non-binary, 21, VIC)* and *'prohibitive rent' (Female, 69, VIC)*. It was well-recognised amongst participants that stable housing was a pre-requisite for good mental health, for example: *'Mental health issues escalate due to the lack of social and affordable housing options. People can't address their mental health issues when homeless' (Female, 54, NSW)* and *'I firmly believe that secure housing is critical to successful recovery pathways for people with mental illness' (Female, 61, ACT)*. The stress associated with rising housing costs appeared to be particularly intertwined in relationships and the need to care and provide for family and loved ones. For this reason, the stress was viewed as particularly concentrated amongst *'families on lower incomes and [with] kids' (Male, 70, SA)*, showing that for many, mental health is contingent on being able to care and provide for family. Another participant described accumulating and competing financial and social needs that emerged throughout the pandemic:

> *'Bills keep coming in, Real Estate Agent asks for deferred rent to be repaid in full in July, daughter needs glasses, other daughter has anxiety and becomes depressed, husband keeps having strange health issues pop up, friends are stressed out, losing business and separating relationships, experiencing restlessness, sleep issues, anger and irritability . . .' (Female, 53, Victoria)*

By contrast, some participants (4.6%, n = 48) reported protective factors for their mental health that emerged as a result of the pandemic, including the Australian Governments' welfare support package for people who experienced reduced available work and income ('JobKeeper'), and the 'coronavirus supplement' for those already receiving government welfare ('JobSeeker')[21], which provided financial stability for participants. For example, one participant described having *'more money in my account then [they] ever had before thanks to JobKeeper'* which was *'good for my mental health' (Female, 39, VIC)*. Another participant described that due to the JobSeeker supplement, *'For the first time in years I was able to pay for essential [medical] treatment' (Female, 27, NSW)*. The additional government financial support also allowed people the freedom to take on other opportunities, with one participant describing being able *'to undertake post grad study'* because they finally had the *'time and financial resources to do so' (Female, 27, VIC)*. One participant who described themselves as having a disability *(Female, 27, NSW)* shared that prior to receiving the supplement *'I felt it would be better to kill myself than try and make it work'* but with the supplement *'For the first time in years money wasn't so tight'*. Another participant *(Female, 41, NSW)* described feeling *'very concerned'* and *'stressed'* about her husband who *'lost the majority of his work through his business'* due to COVID-19 and had *'mental health problems (suicide attempt last year)'*. However, she described how *'. . . with the JobKeeper help from the government, his business is doing okay'*.

Given the benefit that increased financial support had on participants' mental health, it is unsurprising that anticipating the government's reduction or removal of payments had a negative impact on their wellbeing *(n = 28)*. One participant described her mental health as *'up and down'*, and while mostly *'feeling pretty relaxed and cheerful'*, she described getting *'a bit nervous'* with *'JobKeeper going down' (Female, 39, VIC)*. The subsequent reduction in payments of JobSeeker once the coronavirus supplement was removed was described as *'crushing and damaging to your mental health' (Female, 24, TAS)*. One participant described how their already

precarious employment situation was made more stressful through the reduction in JobKeeper:

> *'On top of the reduction in my hours due to covid, I am now sick (not covid-related) and have been off work for several weeks. I am not entitled to sick pay as a sole trader. With JobKeeper payments . . . set to be reduced in the coming months, money is my main worry.' (Female, 37, NSW).*

Not all participants were eligible for JobKeeper or increased JobSeeker payments, and the seemingly inequitable distribution of financial support impacted some participants wellbeing. One participant who worked as a childcare worker was not eligible to receive JobKeeper in mid-2020 and described feeling *'totally used and worthless' (Female, 46, WA)*. The same participant described their sense that the *'government has shown they feel we are essential workers, but the only industry it's not essential to pay'*, showing a perceived mismatch between the increased infection risk for some industries required to work in client-facing conditions, but who are less financially supported than other industries. Other participants reported that, unlike recipients of JobSeeker, recipients of the Disability Support Pension did not receive increases of fortnightly payments. For example, one participant *(Male, 29, NSW)* noted those on the Disability Support Pension are *'getting no help but people on the dole [JobSeeker] [are] getting more money'*, which was in the context of *'stressing about having money to make ends meet'* because of *'the cost of food going up [and] not having money to heat my home in winter.'* This caused the participant to make difficult financial decisions like choosing personally to *'not eat'* in favour of *'making sure my dog is fed'*.

The small amount of baseline welfare payments when the coronavirus supplement was removed was not seen as *'sufficient income to live a 'reasonable life''* especially amongst those with children. Moreover, the return of mutual obligations requirements to receive payments made it difficult; impacting peoples' sense of *'dignity'* when *'the job seeker demands are so high and the payment so small'*. These mutual obligation requirements could be arduous and worsen mental health issues and recovery, as illustrated in the following participant accounts, both of whom have lived experience of mental disorders:

> *'My job network providers were bloody hopeless and 3 times in a row failed to ring me for my phone appointments. Then they kept sending me the same paperwork to sign, so I'd sign it only to receive an email a couple of days later telling me to sign it again. Then the income my partner could earn changed and I kept getting no payments.' (Female aged 48, NSW)*

> *'Social welfare system isn't equipped to support those of us who struggle to work because of mental health issues. I cry every day at my full-time job and would like to focus on recovery, but the tiny rate of Centrelink payments means I keep struggling through'. (Female aged 30, VIC)*

Those who retained work during COVID-19, described the impact it had on their workload: *'people are working longer hours and are more stressed' (Female, 29, NSW)* and a *'work-all-hours mentality' (Female, 43, NSW)* resulting in *'stress from being overworked' (Female, 45, VIC)*. Workplaces could also contribute to poor mental health when cultures and management had a *'lack of support' (Male, 35, NSW)* structures in place to help those experiencing negative mental health outcomes or *'workplaces not understanding how to support colleagues' (Female, 31, NSW)*.

*'Loneliness seems to be a big issue'*: Social and community disconnection. Just over one-in-ten participants (n = 102) described a lack of social and community connection as one of the major issues facing the mental health of Australians during the pandemic, including, *'loneli-ness', 'social fragmentation', 'societal and social exclusion'*. There was a shared sentiment that people were *'feeling lonely and not having connections socially or with their communities' (Female, 40, NSW)* and that *'society has become very individually focused and less about support' (Male, 41, VIC)*. Sometimes this isolation was linked to a lack of physical spaces for socialising: *'[We need] facilities for people and communities to socialise in a healthy environment. Get rid of the poker machines and make pubs a place where people can openly socialise again' (Male, 42, NSW)* and other times this isolation was linked to a cultural shift away from valuing a sense of community: *'I see a lot of people downplaying the importance of socialising as they grow older' (Male, 42, VIC)* and *'We need supportive communities. . .We are too "private" don't share our troubles, don't ask for help, and that's why so many men women and children who are struggling in relationships things have to get really serious before anyone gets help' (Female, 49, NSW)*. Commonly, participants linked the rise of social disconnection to social media. Connections formed on social media were described as *'superficial' (Male, 55, VIC)* and a *'replacement for real connection' (Female, 38, ACT)*.

*Restrictions fragmenting social networks.* Almost one-in-ten participants (n = 82) described the pandemic as exacerbating experiences of isolation, disconnection and loneliness. COVID-19 movement restrictions and isolation measures hampered participants' ability to connect with loved ones, leading to *'separation'* and *'disconnection'* from friends and family, including those from interstate and overseas, leading to feelings of *'loneliness', 'grief'* and *'loss'*. For example, one participant shared *'COVID has affected my mental health, because it has separated me completely from my family and friends who live in Sydney. I moved to Melbourne. . . [knowing] I could get on a plane at any time to visit my family. It felt extremely strange to have that option taken away from me' (Female, 26, Victoria)*. People felt the impacts on social contact strongly, with one participant describing: *'The restrictions on social contact, while understandable and necessary are draining. Likewise mask-wearing is necessary, but it means we don't see people smiling when we used to.'* (Female 46, VIC). Moreover, this social dislocation was all-encompassing, seeping into multiple aspects of participants' social roles and identities, as the following account reflects:

> *'Being single, the option of dating was eliminated. As a friend, the opportunity to connect with my nearest and dearest was altered. As an employee, I felt disconnected from my work and my colleagues.' (Female, 25, NSW)*

Isolation due to restrictions prevented participants from getting necessary social support from their network when experiencing psychological hardship: *'the impact of this pandemic is relentless, and we can't even meet up with friends for coffee' (Female, 53, Victoria)* and *'Isolation led to losing opportunities to intervene. I felt disconnection and didn't know how to help others.' (Female, 51, NSW)*. Broadly, participant accounts described a less cohesive Australian society as a result of COVID-19, for example *'we are undergoing major social change due to covid which is making people scared and distrustful' (Female, 57, NSW)*.

Even 12 months after the start of the pandemic, participants reported how the break in social contact had a longstanding effect on their feelings about socialising, with 10 participants explicitly describing new feelings of *'agoraphobia'*. The following accounts, from participants both with and without mental disorders, shine further light on feelings of discomfort socialis-ing again in a period where restrictions had eased nationally:

*'I feel much more emotionally fragile now. I also feel more socially anxious—being around a lot of people doesn't feel normal anymore.' (Male, 32, VIC)*

*'My anxiety is severe, and most situations stress me out so badly I'm on edge and throw up. I actually have to make the effort to communicate with my friends and family online. I had a panic attack last week and couldn't attend when I was supposed to attend my first in person class since March 2020.' (Female, 23, VIC).*

Not everyone felt disconnected: some enjoyed the more 'localised' nature of their world and social connections: *'One positive thing COVID did do for me is make my world more local and help me slow down. I don't want to lose this.' (Female, 32, NSW).*

**Inaccessible or inappropriate services and stigma: How COVID-19 exacerbated barriers to mental healthcare.**   More than one in five participant responses (n = 226) highlighted that COVID-19 increased pressures on an already over-burdened mental health service system. They identified three main barriers to improving their mental health, each of which was exacerbated during COVID-19: the confusing, expensive and low-capacity nature of the current mental health service system; holes in the current care available through the service system; and absent or stigmatising rhetoric around mental health issues and social inequity, which were present on a political and societal level.

*Inaccessible mental health services.* Many participants (n = 90) described mental health treatment, services and resources as *'prohibitively expensive'* and *'unaffordable. . .even with Medicare rebates.'* Some participants felt that the cost of treatment could contribute to mental ill-health. The following accounts from individuals with lived experience of differing mental disorders illustrate this finding:

*'While I acknowledge the government has increased the number of Medicare subsidised visits, the out-of-pocket expense makes receiving regular, effective psychological treatment prohibitive especially as a single parent who is trying to work and support a young family.' (Female, 37, NSW).*

*'Given that the majority of people suffering from severe mental health issues are low income earning, this leaves many people in crisis situations with no relief.' (Female, 29, VIC).*

*'It's hard because I am already stressed about finances, but seeing a mental health professional costs money, which compounds the stress' (Female, 25, NSW).*

Participants (n = 47) also described long waitlists as a barrier to accessing needed psychology treatment:

*'When people are in crisis, they need the help at that time. Not six months down the track when an opening finally becomes available at the counselling centre.' (Non-binary person, 70, TAS).*

*'There is no point making an appointment in a month if someone needs help immediately'. (Female, 60, NSW).*

Long waitlists could negatively impact continuity of care and therapist-patient rapport. Long wait-times meant participants had *'been unable to see the same provider more than once or twice.'* Being *'seen by different [service providers] all the time'* made it difficult *'to create any trust'* between clinician and client.

There was also discussion around the lack of mental health services and support (n = 57), and this appeared to be particularly concentrated amongst participants in rural, regional, and remote regions of Australia. One participant *(Female, 25, VIC)* from *'a large town'* in regional Victoria described the available care in their state: *'there is only one bulk billing psychiatrist in my area and none in the nearby two large towns.'* Similarly, another participant described their experience accessing help in their regional location as *'pathetic'*, with *'no [mental health] services within 1100 kms'* of where they lived. One participant described that even where they lived in *'Western Sydney'* there was *'very limited psychiatric services. . . leading to very long wait-lists, poor service, or no access'*.

*Inappropriate mental health services.* Overburdened mental health services and long wait-lists during the pandemic meant some participants (n = 36) described feeling they had little choice over which clinician would provide the *'right'* fit with their care. One participant described *'the shortage of counselling and the short duration of medical referral. . .exacerbated by the pandemic'* was a barrier *to 'obtaining help from the right person not just anyone who is available' (Female, 63, VIC)*. This was particularly the case amongst people in rural/regional areas where participants described being *'unable to find any psychologists with availability in [their] local area'*, particularly bulk billed services. One participant described how the low *'availability of suitable services'* meant *'too many people get palmed off to services who are not equipped to help them and no one listens' (Female, 54, NSW)*.

*Holes in the service system.* Participant accounts (n = 38) highlighted gaps within the current mental health treatment system for people whose mental health needs are not yet acute. Falling within this 'gap' could have serious ramifications for a person's access to appropriate and timely treatment. For example, one participant, whose *'nine-year-old has been diagnosed with anxiety and panic attacks'* was not perceived as *'bad enough to see anyone for at least 3 months'*. Her child's enforced wait in accessing treatment meant being unable to intervene or prevent worsening of his symptoms: *'Massive underfunding means people can't access support pre-emptively, or even in a timely fashion once things get bad.'* Another participant shared this sentiment: *'There is little support for people with chronic and debilitating mental health problems who aren't experiencing severe enough (i.e., outwardly noticeable) symptoms to require being held against their will'. (Female, 27, NSW)*. Moreover, the lack of intensive ongoing support provided to people following crisis care was described by one participant as *'almost Band-Aid treatment'*.

*Societal barriers to getting better.* Many participants (n = 63) described the lack of recognition and acknowledgment of mental health problems within Australian leadership, for example *'People, especially politicians, underplay the seriousness of mental health' (Male, 35, TAS)*. This *'limited'* government recognition that *'mental health is critical to the health and well-being of our nation as a whole'* impacted the experiences of Australians by *'making the stigma worse'* around mental health.

According to participants (n = 113), mental health issues were described as *'still very taboo'* within Australian society *(Female, 26, NSW)*. There was a collective sentiment that increased awareness building about *'mild'* mental health issues did not translate into social *'acceptability'* of talking about experiences of mental health problems:

> *'It's all very well asking 'RUOK', [but] when some one says 'no' people are not really interested in your problems. . . 'I am crying writing this as the loneliness is overwhelming'. (Female, 63, NSW)*.

> *'People just roll their eyes, as if they think that people who have mental health issues are weak, need to 'pull themselves together'. (Female, 67, VIC)*.

Societal conceptualisations of what constitutes *'acceptable'* mental ill-health reflected a *'complete lack of understanding of what mental health looks like'*. As a result, some participant accounts reflected a sense of *'un-speakability'* of mental health issues publicly, describing *'the difficulty in having the language to communicate what is happening and to be able to hear what people are saying.' (Female, 65, VIC).*

Several participants (n = 13) with lived experience of mental disorders described that mental health public awareness campaigns place the burden on individuals experiencing mental health issues to find their own solutions:

> *'Mental health messaging encourages people to reach out for help if they need it, which is great, but it often places the burden of taking action on an individual at a time when they're most vulnerable and least able to take that action.' (Female, 40, QLD)*

Instead, participants noted the onus of supporting and facilitating access to treatment should be shared amongst a person's community:

> *'I think it would be helpful if there was more education for people on offering 'mental health first aid' and especially practical support to a family or friend who discloses a mental health situation.' (Female, 40, QLD)*

The stigma around *'getting help'* and the widespread conceptualisation of mental disorder as an individual responsibility were described as preventing people from seeking support. Participants described how such experiences led them to try to cope alone until eventually *'they reach a point where they cannot go on without some help.'* One young participant reflected *'It takes a lot of courage to speak with someone regarding your mental health. . .I found it a bit overwhelming going to my GP with my mental health issue' (Female, 22, NSW).* She recommended *'making mental health [services] more accessible'* by de-pathologising mental health and *'trying to eliminate'* the idea *'that mental health is a medical problem'*. One participant described the help they eventually received after overcoming barriers to access as *'the best thing I could have done.' (Female, 33, NSW).*

## Discussion

Australian community members are uniquely placed to provide advice and guidance around their mental health needs and experiences during the COVID-19 pandemic and beyond. Existing Australian research on mental health priorities has focussed on the perspectives of consumers of the mental health service system and their carers, often to the exclusion of community members not in contact with the service system. Given the society-wide disruption of the pandemic on Australians, our study provided a necessarily broad viewpoint in understanding the main issues and barriers faced by participants in achieving good mental health going forward.

The psychological discipline is often critiqued for its reliance on biomedical or deficit models of mental health, which characterise poor mental health as an individual 'problem' and disregard the impact a person's social and environmental context has on their emotional wellbeing [22]. Our participant accounts instead reflected a biopsychosocial and social determinants model of mental health: participants attributed experiences of poor mental health to a person's socioeconomic context, including low income, a lack of meaningful employment and absent or low-quality relationships [17,23]. They also described cultural determinants of poor mental health, including widespread stigmatising political and community rhetoric about what it means to be unemployed, receive welfare payments or to require mental health

treatment, which degraded the wellbeing of those in these categories. Finally, participants identified how flaws in institutional policies designed to support Australians experiencing hardship and ill-health, such as the expensive, difficult to navigate, low-capacity mental health service system and mutual obligation requirements of the welfare system, were recognized as having a contradictorily negative effect on recipients' mental health and their ability to recover.

Taken together, these findings reveal that many Australian community members recognise that social and economic factors play an integral role not only for their own mental health, but also in the broader picture of mental health and wellbeing of all Australians. Moreover, findings from the present study suggest that a substantial number of Australian community members demonstrate sophisticated, place-based knowledge and understanding of the mental health system, and where it currently *does* and *does not* meet their needs.

### The pandemic and mental health

Findings demonstrated that the COVID-19 pandemic 'pressurised' existing triggers for poor mental health by amplifying financial stress and reducing social support and connection. The associated increased demand on the mental health service system, which was already over-burdened prior to the pandemic [24], proved a further obstacle to recovery by restricting access to critical treatment and care.

Participant accounts provided richer insights into findings that have been identified via quantitative studies examining the COVID-19 pandemic and mental health. Namely, COVID-related disruption to work and social functioning has been linked with the changing mental health of many Australians; elevating symptoms of depression and anxiety [25] and increasing risk of stress, loneliness, depression, anxiety and self-harm [26–28].

This study showed that a major pathway through which the pandemic impacted Australians' mental health is through the disruption to their social functioning: the ability to engage in regular social activity, maintain social connectedness, and fulfil social roles and identities in families, relationships, work, and other social activities [29]. The major disruption the pandemic played in participants' social worlds was their ability to connect with loved ones, give and receive support and care and maintain social roles and identities that had provided a meaningful structure to their lives prior to the pandemic. Unfortunately, this social disruption seemed likely to extend beyond COVID-19 restrictions with many participants reporting heightened anxiety about re-entering their social world post-pandemic, which sometimes prevented them from reverting to their usual social engagements. Interestingly, participants also couched the mental health impacts of lost income around the disruption this had on their social functioning and roles: in particular, their need to support and care for family by paying for essential costs. Loss of employment was also positioned as a loss to their meaningful pre-pandemic social identity and status.

Our finding that the pandemic disrupted participants social relationships and social identities is concerning, given evidence demonstrating that a person is more likely to experience post-traumatic stress when the traumatic event has undermined their valued social identities, like a work- or friendship-based identity, and more likely to be resilient to adverse psychological outcomes when their valued social identities can be maintained or redefined through positive social connections [30]. Unfortunately, COVID-19 restrictions, including social distancing, lockdowns and changes to employment and financial loss has meant many Australians may leave the pandemic feeling socially disconnected with lost valued identities, rather than connection and support.

Participants also recognised that the pandemic has compounded existing barriers to receiving timely and appropriate mental health treatment, including the inaccessible, unsuitable,

and unaffordable nature of mental health services and the broad stigmatisation of mental illness. The rapid increase in demand for Medicare-funded mental health services and crisis support helplines during the pandemic [15,31] has stretched an already overloaded mental health service sector and exacerbated system-level issues, such as long wait times and lack of choice around treatment. As a result, access to mental health services has been rendered more difficult for mental health consumers, particularly for those living in rural and remote areas and the socioeconomically disadvantaged, who are more likely to report reduced engagement with psychosocial services due to pandemic-induced disruptions to service provision [10]. Barriers to mental health care, such as delays to treatment and public stigma, hinder access and engagement with mental health services, and this has been associated with long-term negative consequences, including worse mental health outcomes, poor quality of life and impaired social functioning [32]. There is an urgent need for uninterrupted access to mental health supports with steps taken to actively reduce barriers to mental healthcare during the pandemic, as recognised by the World Health Organisation [33].

## Implications for policy and practice

Insights gained from the present study provide an opportunity for policymakers to draw on the expertise of Australians' lived experience and address the core priority issues raised in participant accounts. Our findings show that mental healthcare is not just about psychiatric service provision but is also financial support and social support. Therefore, whole-of-government policies spanning social services/welfare, finance, housing and the built environment, education, family and community and workforce are needed to achieve tangible impacts on Australians' mental health.

Many of the barriers of the service system described by participants were not unique to the pandemic context and have been detailed at length in the Productivity Commission report [24]. However, these accounts give voice to the lived frustration, stress and disappointment of those who try to negotiate appropriate care. While COVID has increased public discourse around mental health issues, it has also escalated urgency for better models of service provision and care. Tackling these priority issues could involve improving mental health service capacity, design and geographic distribution, delivery and access to meet increasing demand during the pandemic; increasing prevention and early intervention strategies; developing stigma reduction interventions and targeting the social determinants of mental health by adopting a more integrated framework across different service sectors [24]. Policymakers should also consider existing social inequalities and equity-based implementation processes in policy planning to ensure all Australians benefit from mental health reform.

Our findings also suggest that Australians' mental health may be improved and safeguarded post-pandemic through the development of initiatives that reconnect Australians to meaningful social identities, such as employment, study and training opportunities. There is also a need for government strategies that re-connect Australians to meaningful relationships and communities. Examples include grant schemes to develop or maintain community groups, such as sports teams, 'men's sheds', or choirs, that can help build community belonging, social cohesion and collective resilience, which has been shown to improve participants' mental health and withstand future crises or stressful events [34,35].

Participants shared concerns around the lack of recognition and acknowledgment of mental health within Australian leadership and politics, particularly during the pandemic, and how this has contributed to existing societal stigmatisation and discrimination against those with mental illness. Our findings suggest further work is required to increase government and community discourse around mental health. In recent months, there has been greater public

dialogue and awareness of pandemic-related mental health issues, and the importance of seeking mental health support, including a substantial 'mental health' federal budget [36]. We can leverage this recent focus as an opportunity for a cultural shift in the public rhetoric around mental disorders. Existing media guidelines including MindFrame already exist which present sensitive, destigmatising and appropriate ways of discussing mental health issues [37]. Guidelines like these could be used to frame political speeches and policy announcements. Our findings also point to clear implications for welfare provision in Australia, including a need to destigmatise what it means to be unemployed and the removal of mutual obligation requirements in welfare schemes, which government reports have already acknowledged can have detrimental effects on mental health [24].

Overall, these findings are a timely reminder that the voices, stories and perspectives of the Australian population have an important role in shaping and informing our national response to mental health and wellbeing, both during and beyond COVID-19. We must continue to amplify and engage the voices of lived and living experiences in decision-making and co-design processes around mental health to create meaningful change; change that values, respects, listens and responds to the people and communities it serves [5].

## Strengths and limitations

This study is the first of its kind and summarised the complex mental health needs and priorities of Australians within the COVID-19 context. Despite clear strengths, the current study is not without some limitations. The study relied on convenience sampling, potentially including community members who are more likely to be outspoken about mental health issues and educated enough to participate in an online university-led survey. Consequently, while the participant accounts provide a broad range of experiences, the findings are not representative or generalisable across all Australians. The current sample was primarily comprised of female participants, meaning an under-representation of male voices. Although beyond the scope of the current study, this research could have been improved by targeting recruitment of specific populations at increased risk of COVID-19 infection and the psychological fallout of restrictions, such as youth, rural populations or specific cultural groups such as the Aboriginal and Torres Strait Islander population [38]. Despite these limitations, these qualitative findings offer deeper insights into the experiences and stories of many Australians, with and without lived experience, and this nuance can pave the way for clear policy and practice recommendations. This study lies at the forefront of the conversation around the mental health priorities and needs of Australians during the COVID-19 pandemic. In order to capture the mental health concerns of Australians, we have necessarily asked broad, open-ended survey questions. Future research would benefit from further exploring issues surrounding financial stability, social and community connection, and barriers to accessing mental healthcare, to better understand the risk and protective factors impacting Australians' mental health.

## Conclusions

Participants highlighted that economic instability and social isolation were drivers of poor mental health amongst many in Australia, and both experiences were heightened and exacerbated by the pandemic. Australians who are unwell face multiple barriers to seeking and receiving good mental health care, including an already over-burdened mental health service system and widespread stigma around mental health and help-seeking. This is a critical opportunity for policymakers and health practitioners to draw on the expertise of Australians' lived experience, address priority issues of economic and social dislocation through targeted policy

planning, and ultimately build a more responsive, integrated and effective mental health system, during the COVID-19 pandemic and beyond.

## Acknowledgments

This work was undertaken as part of Australia's Mental Health Think Tank. The authors acknowledge the Alone Together participants who very generously shared their stories with us and the housing and homelessness services who supported the research, including the Haymarket Foundation, Wayside Chapel, the Exodus Foundation and Hope Street, amongst others.

## Author Contributions

**Conceptualization:** Marlee Bower, Scarlett Smout, Emma Barrett, Maree Teesson.

**Data curation:** Marlee Bower, Amarina Donohoe-Bales, Scarlett Smout, Julia Boyle.

**Formal analysis:** Marlee Bower, Amarina Donohoe-Bales.

**Funding acquisition:** Marlee Bower, Scarlett Smout, Maree Teesson.

**Investigation:** Marlee Bower, Amarina Donohoe-Bales, Scarlett Smout, Andre Quan Ho Ngyuen, Julia Boyle, Emma Barrett, Maree Teesson.

**Methodology:** Marlee Bower, Amarina Donohoe-Bales, Julia Boyle, Maree Teesson.

**Project administration:** Marlee Bower, Amarina Donohoe-Bales, Julia Boyle.

**Resources:** Marlee Bower, Amarina Donohoe-Bales, Scarlett Smout.

**Software:** Marlee Bower, Amarina Donohoe-Bales.

**Supervision:** Marlee Bower, Emma Barrett, Maree Teesson.

**Validation:** Marlee Bower.

**Visualization:** Marlee Bower.

**Writing – original draft:** Marlee Bower, Amarina Donohoe-Bales, Scarlett Smout, Andre Quan Ho Ngyuen, Julia Boyle, Emma Barrett.

**Writing – review & editing:** Marlee Bower, Amarina Donohoe-Bales, Scarlett Smout, Julia Boyle, Emma Barrett, Maree Teesson.

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
