## [Decision Letter · Decision Letter 0]

10 Feb 2022

PONE-D-21-30862In their own words: Understanding Australians’ priority concerns regarding mental health in the context of COVID-19PLOS ONE

Dear Dr. Bower,

Thank you for submitting your manuscript to PLOS ONE. After careful consideration, we feel that it has merit but does not fully meet PLOS ONE’s publication criteria as it currently stands. Therefore, we invite you to submit a revised version of the manuscript that addresses the points raised during the review process.

We look forward to receiving your revised manuscript.

Kind regards,

Professor Benjamin Tan, BNSc MMed PhD RN

Journal Requirements:

Reviewers' comments:

Reviewer's Responses to Questions

**Comments to the Author**

1. Is the manuscript technically sound, and do the data support the conclusions?

Reviewer #1: No

Reviewer #2: Yes

2. Has the statistical analysis been performed appropriately and rigorously? 

Reviewer #1: No

Reviewer #2: N/A

3. Have the authors made all data underlying the findings in their manuscript fully available?

Reviewer #1: No

Reviewer #2: Yes

4. Is the manuscript presented in an intelligible fashion and written in standard English?

Reviewer #1: Yes

Reviewer #2: Yes

5. Review Comments to the Author

Reviewer #1: The manuscript with the title "Understanding Australians’ priority concerns regarding mental health in the context of COVID-19" describes the findings from a qualitative survey conducted to assess the impact of COVID-19. The manuscript is interesting, however, for a scientific paper, the overall quality of the manuscript is poor and does not rich the PONE standard for publication. The manuscript requires a significant revision before it can be published.

Specific comments

Abstract: The sentence "Of the 1,350 participants who completed the first follow-up survey, a total of 1,037 participants, who ranged in sex (69.9% female), age (M = 40-49 years), state/territory of residence, and socioeconomic status, shared responses to two open-ended questions regarding the most important issues for mental health in Australia and the impact of COVID-19 on their individual mental health." is convoluted and should be rephrased.

Introductions: The objectives of the study should be clearly highlighted, and how the study addresses them by stating the hypotheses proposed. Further, it is not clear what the knowledge gap is apart from that this study being made from a qualitative research. Please state what is new that this study addresses which the previous study failed to pick up in their design. The authors should review the available literature and identify the knowledge gap.

Method section: More details should be added in the method section, covering a full ethics statement, a justification for choosing only three cities for the study, and data quality control procedure.

Results: The result section should be rewritten to include the associated statistics instead of describing the results in an abstract (we readers would like to see the accompanying data for the claims you make in the text), i.e., align the text to the data/results which can be verified. Moreover, most of the results reported are based on responses from one or very few respondent(s). Is this representative of the greater Australian population as the manuscript title claim? No! Thus, these results are of low quality and deep discussion and/or making strong inferences from, should be avoided. Further, it is also recommended that the result should be analysed for the different geographic locations.

Discussion: The discussion should be revised to ensure it is supported by the data.

The manuscript title should be changed. The results reported are based on few individual responses and thus not representative of the greater Australian population. Hence, using "Australians" is not warranted.

Reviewer #2: This paper outlined the experiences of many adult Australians during the COVID-19 pandemic, including their thoughts about mental health. It was written beautifully, and was conceptually sound. I have very few comments. I wish the authors well in their work.

Introduction: This section was clear, and rationale was sound.

Methods: Could you please give an example after "support service organisations" as it was unclear to me when I was reading this sentence. Page 6.

Results

- Page 10: 'participant accounts' should be 'participants' accounts'

- some quotes are italicized and some are not. Perhaps, change for consistency. e.g., Page 12.

- Narrative approach was helpful and clear.

Discussion

- Well written and no changes needed.

6. PLOS authors have the option to publish the peer review history of their article (what does this mean?). If published, this will include your full peer review and any attached files.

Reviewer #1: No

Reviewer #2: No

---

## [Author Response · Author response to Decision Letter 0]

7 Apr 2022

We would like to thank the reviewers for taking the time to review our manuscript and for their thoughtful and insightful advice. We hope that our edits and responses below satisfactorily address any issues or concerns raised in this process. Thank you for the opportunity to strengthen our manuscript.

Overarching Feedback

1. Is the manuscript technically sound, and do the data support the conclusions?

Reviewer #1: No

Reviewer #2: Yes

Author response: Our paper has adopted scientifically rigorous qualitative analysis and synthesis methodologies. As such, the manuscript does not contain experimental control groups, replication procedures or quantitative research practices. Consistent with Reviewer #2’s response, this manuscript adheres to qualitative research guidelines and best practice by specifying the chosen qualitative approach; describing how and why participants were recruited; documenting participant consent, sample sizes and ethics approval; detailing qualitative data collection, analysis and processing methods; including participant quotes; and using findings to draw appropriate conclusions and implications. While we believe that our findings strongly support the conclusions made, we have made amendments to improve clarity and connections between the results and discussion section (see Reviewer’s comments below).

2. Has the statistical analysis been performed appropriately and rigorously?

Reviewer #1: No

Reviewer #2: N/A

Author response: Given the qualitative nature of this paper, the manuscript does not include quantitative statistical analyses. Therefore, this question is not applicable, in line with Reviewer #2’s response. 

3. Have the authors made all data underlying the findings in their manuscript fully available?

Reviewer #1: No

Reviewer #2: Yes

Author response: The data have not been made publicly available in order to maintain participant confidentiality and privacy. While de-identified, participant responses contain sensitive and confidential information and therefore cannot be shared publicly. Data can be made available upon reasonable request to the corresponding author. 

4. Is the manuscript presented in an intelligible fashion and written in standard English?

Reviewer #1: Yes

Reviewer #2: Yes

Author response: N/A 

Reviewers’ Feedback

Reviewer #1

The manuscript with the title "Understanding Australians’ priority concerns regarding mental health in the context of COVID-19" describes the findings from a qualitative survey conducted to assess the impact of COVID-19. The manuscript is interesting, however, for a scientific paper, the overall quality of the manuscript is poor and does not rich the PONE standard for publication. The manuscript requires a significant revision before it can be published.

Specific comments

Abstract: The sentence "Of the 1,350 participants who completed the first follow-up survey, a total of 1,037 participants, who ranged in sex (69.9% female), age (M = 40-49 years), state/territory of residence, and socioeconomic status, shared responses to two open-ended questions regarding the most important issues for mental health in Australia and the impact of COVID-19 on their individual mental health." is convoluted and should be rephrased.

Author response: We agree the sentence was convoluted. We have rephrased to be clearer and more concise. 

“A total of 1,037 participants, ranging in sex (69.9% female), age (M = 40-49 years), state/territory of residence, and socioeconomic status, shared responses to two open-ended questions in the first follow up survey regarding mental health experiences and priorities during COVID-19.”

Introductions: The objectives of the study should be clearly highlighted, and how the study addresses them by stating the hypotheses proposed. Further, it is not clear what the knowledge gap is apart from that this study being made from a qualitative research. Please state what is new that this study addresses which the previous study failed to pick up in their design. The authors should review the available literature and identify the knowledge gap.

Author response: Thank you for your comments and highlighting this gap. We have amended the introduction to ensure all knowledge gaps are clearer to the reader. We have also included aims and hypotheses, as per the Reviewer’s suggestions.

Method section: More details should be added in the method section, covering a full ethics statement, a justification for choosing only three cities for the study, and data quality control procedure.

Author response – We have made the following changes in line with the Reviewer’s suggestions:

1. Included further detail about the consent procedure utilised.

“An active consent procedure was used. Participants were provided with a participant information statement and then asked to confirm their consent before they were able to commence the questionnaire. In addition, the steps to withdraw from the study at any time were clearly explained. Prior to commencing each follow-up questionnaire, participants were also asked to actively re-confirm their consent to take part in the study.”

2. Participants from all Australian states and territories were recruited and included in our study, not just three cities as per Reviewer #1’s comments. We have made this clearer in the abstract, method and results section. 

3. We have reiterated steps in our methodology section to highlight strategies used to maintain quality control undertaken during data coding, analysis and interpretation.

Results: The result section should be rewritten to include the associated statistics instead of describing the results in an abstract (we readers would like to see the accompanying data for the claims you make in the text), i.e., align the text to the data/results which can be verified. Moreover, most of the results reported are based on responses from one or very few respondent(s). Is this representative of the greater Australian population as the manuscript title claim? No! Thus, these results are of low quality and deep discussion and/or making strong inferences from, should be avoided. Further, it is also recommended that the result should be analysed for the different geographic locations.

Author response – We have made the following changes in line with the Reviewer’s suggestions:

1. The number and percentage of total participant responses for each theme/finding has been added to the manuscript, as per the Reviewer’s suggestion, to provide greater context and transparency in reporting.

2. In response to Reviewer #1’s comment “most of the results reported are based on responses from one or very few respondent(s)”: we wish to clarify that our results and themes are based on multiple participant responses, not singular responses. Due to limitations with word count, we have selected exemplary participant quotes that capture broader themes to make arguments. Participant quotes have been dispersed throughout the paper to illustrate broader findings and support our overarching claims. We have included a sentence in the results section to explain the use of de-identified quotes. The use of associated statistics (first point) will hopefully clarify this further. 

3. As this is a primarily qualitative study, it is not convention and not practicable to analyse by different geographic locations for thematic analysis (i.e., it would require saying, ‘this theme emerged in participants from QLD, WA, NSW and NT’, implying that this was not relevant for participants from other areas, which is inaccurate for qualitative research where participants aren’t required to endorse/not endorse criteria). Where individual participant quotes are included, the State of residence has been included. 

Discussion: The discussion should be revised to ensure it is supported by the data.

Author response: Thank you for alerting us to this issue. Alterations have been made to the discussion section to soften the language used to describe results to avoid generalisations to broader Australians. For example, the term ‘Australians’ was changed to ‘participants’ or ‘some Australians’ or ‘Australian community members’.

The manuscript title should be changed. The results reported are based on few individual responses and thus not representative of the greater Australian population. Hence, using "Australians" is not warranted.

Author response: We have now changed the manuscript title to“In their own words: An Australian community sample’s priority concerns regarding mental health in the context of COVID-19”

Reviewer #2

This paper outlined the experiences of many adult Australians during the COVID-19 pandemic, including their thoughts about mental health. It was written beautifully, and was conceptually sound. I have very few comments. I wish the authors well in their work.

Specific comments

Introduction: This section was clear, and rationale was sound.

 Author response: N/A

Methods: Could you please give an example after "support service organisations" as it was unclear to me when I was reading this sentence. Page 6.

Author response: Corrected and provided more detail. 

Results

- Page 10: 'participant accounts' should be 'participants' accounts'

 Author response: Corrected.

- Some quotes are italicized and some are not. Perhaps, change for consistency. e.g., Page 12.

Author response: Corrected. All quotes and demographic details have been italicised for consistency.

- Narrative approach was helpful and clear.

 Author response: N/A

Discussion

- Well written and no changes needed.

 Author response: N/A

---

## [Editor Report · Decision Letter 1]

10 May 2022

In their own words: An Australian community sample's priority concerns regarding mental health in the context of COVID-19.

PONE-D-21-30862R1

Dear Dr. Bower,

We’re pleased to inform you that your manuscript has been judged scientifically suitable for publication and will be formally accepted for publication once it meets all outstanding technical requirements.

Kind regards,

Professor Benjamin Tan, BNSc MMed PhD RN

Additional Editor Comments (optional):

The reviewers' comments have been well addressed in the revised manuscript. The academic editor is satisfied with your revisions. A decision has therefore been made to accept your paper for publication.

---

## [Editor Report · Acceptance letter]

12 May 2022

PONE-D-21-30862R1 

In their own words: An Australian community sample’s priority concerns regarding mental health in the context of COVID-19. 

Dear Dr. Bower:

I'm pleased to inform you that your manuscript has been deemed suitable for publication in PLOS ONE. Congratulations! Your manuscript is now with our production department. 

Kind regards, 

on behalf of

Professor Benjamin Tan 

Academic Editor

PLOS ONE